# Therapeutic Efficacy of YM155 to Regulate an Epigenetic Enzyme in Major Subtypes of RCC

**DOI:** 10.3390/ijms25010216

**Published:** 2023-12-22

**Authors:** Seong Hwi Hong, Young Ju Lee, Eun Bi Jang, Hyun Ji Hwang, Eun Song Kim, Da Hyeon Son, Sung Yul Park, Hong Sang Moon, Young Eun Yoon

**Affiliations:** 1Department of Urology, Hanyang University College of Medicine, Seoul 04763, Republic of Korea; hshshshsh90@hanyang.ac.kr (S.H.H.); younglee22@hanyang.ac.kr (Y.J.L.); syparkuro@hanyang.ac.kr (S.Y.P.); moonuro@hanyang.ac.kr (H.S.M.); 2Department of Translational Medicine, Hanyang University Graduate School of Biomedical Science & Engineering, Seoul 04763, Republic of Korea; oneleaf826@naver.com (E.B.J.); hjamelia@hanyang.ac.kr (H.J.H.); thaql11@hanyang.ac.kr (E.S.K.); dlwlrma88@hanyang.ac.kr (D.H.S.)

**Keywords:** renal cell carcinoma, BIRC5, YM155, histone acetylation

## Abstract

Renal cell carcinoma (RCC) is the most common type of kidney cancer and includes more than 10 subtypes. Compared to the intensively investigated clear cell RCC (ccRCC), the underlying mechanisms and treatment options of other subtypes, including papillary RCC (pRCC) and chromogenic RCC (chRCC), are limited. In this study, we analyzed the public databases for ccRCC, pRCC, and chRCC and found that BIRC5 was commonly overexpressed in a large cohort of pRCC and chRCC patients as well as ccRCC and was closely related to the progression of RCCs. We investigated the potential of BIRC5 as a therapeutic target for these three types of RCCs. Loss and gain of function studies showed the critical role of BIRC5 in cancer growth. YM155, a BIRC5 inhibitor, induced a potent tumor-suppressive effect in the three types of RCC cells and xenograft models. To determine the mechanism underlying the anti-tumor effects of YM155, we examined epigenetic modifications in the BIRC5 promoter and found that histone H3 lysine 27 acetylation (H3K27Ac) was highly enriched on the promoter region of BIRC5. Chromatin-immunoprecipitation analysis revealed that H3K27Ac enrichment was significantly decreased by YM155. Immunohistochemistry of xenografted tissue showed that overexpression of BIRC5 plays an important role in malignancy in RCC. Furthermore, high expression of P300 was significantly associated with the progression of RCC. Our findings demonstrate the P300-H3K27Ac-BIRC5 cascade in three types of RCC and provide a therapeutic path for future research on RCC.

## 1. Introduction

Renal cell carcinoma (RCC), which accounts for more than 90% of all kidney cancers, is one of the most lethal malignancies [1,2]. RCC is classified into 20 types, and the most common are clear cell RCC (ccRCC; 75%), papillary RCC (pRCC; 10%), and chromophobe RCC (chRCC; 5%) [3]. Most of the research on kidney cancer has mostly focused on ccRCC. According to the National Comprehensive Cancer Network (NCCN) guidelines, the recommended current standard treatments for pRCC and chRCC are not different from those for ccRCC because there have been no specific medications designed for pRCC or chRCC, and they recommend participation in clinical trials [4]. The incidence of RCC is increasing every year and effective targets of RCC are limited; there are also practical difficulties in treating various non-ccRCCs. Early diagnosis of RCC is difficult because there are no symptoms or representative markers; thus, without accidental finding of renal mass on abdominal image work up, RCC is easily missed. Additionally, the 5-year survival rate is very low once metastatic disease is diagnosed [5,6]. The standard treatment for organ-confined RCC is surgical excision; however, metastasis and recurrence are frequent, and RCC is resistant to most therapy such as conventional chemotherapy, radiation therapy, and hormone therapy [7,8,9,10,11]. Sunitinib, a tyrosine kinase inhibitor that blocks growth of blood vessels, has been used as a first-line treatment for RCC for over 10 years, but it only inhibits angiogenesis without effects on the tumor itself; therefore, it only extends the lifespan by several months [12]. Recently, immune therapeutics, such as nivolumab and ipilimumab, are being used in RCC treatment; however, little is known of the efficacy of these medications, and they have not been recommended for pRCC and chRCC [13]. Therefore, a novel effective treatment for all major subtypes of RCC, including ccRCC, pRCC, and chRCC, is urgently needed.

Inhibitor of apoptosis (IAP) is a highly conserved gene family present in most organisms in which the encoded proteins suppress apoptosis by regulating caspase activity [11,14,15]. The IAP family contains eight proteins, BIRC1, BIRC2, BIRC3, BIRC4, BIRC5, BIRC6, BIRC7, and BIRC8, according to the composition of approximately 8 domains BIR, NACHT, UBA, CARD, RING, Coiled-Coil, UBC, and LRR [11,16]. BIRC5, also known as survivin, inhibits caspase-3 and -7 and interferes with caspase-9 activity, thereby suppressing the release of cytochrome c from mitochondria [16,17,18]. BIRC5 is significantly upregulated in most tumors, and its dysregulation affects cancer aggressiveness, recurrence, drug resistance, and patient survival [19,20,21]. The induction of apoptosis via inhibition of BIRC5 is a promising target for the treatment of various cancers [22,23,24,25,26,27]. However, limited studies have been performed in RCC.

In this study, we explored potential therapeutic targets that can induce cell death of ccRCC, pRCC, and chRCC by performing IAP expression analysis and we investigated the underlying mechanism of a potential drug aimed at the novel target.

## 2. Results

### 2.1. Correlation between Overexpressed BIRC5 and RCC Tumor Progression

To find a potential therapeutic target for the three main subtypes of RCCs, we analyzed TCGA data for ccRCC, pRCC, and chRCC (Figure 1). We examined genes that changed by 2-fold or more in all three RCCs and identified 1118 genes that were low-expressed and 191 genes were high-expressed. With the aim of identifying a target that may induce death of cancer cells, we selected transcription factors related to apoptosis from the genes overexpressed in RCC. We selected BIRC5 as BIRC5 was significantly increased by more than 2-fold in tumors compared with non-tumor samples; similar patterns were observed in paired analysis of the 3 major RCC subtypes (Figure 1a,b). We then analyzed the mRNA expression of the eight members of the BIRC family and found that only BIRC5 and BIRC7 were overexpressed in tumor tissues (Figure 1c). Similar to BIRC5, BIRC7 was significantly overexpressed in tumors, and a significant difference was also found in the paired analysis (Figure 1d,f).

To examine the relationship between cancer development and BIRC5 and BIRC7, we compared the expression levels in patients at various stages. In TCGA data for three RCCs, high stages (stage III and IV) were associated with increased expression of BIRC5 (Figure 1f). Thus, BIRC5 was found to have a close correlation with cancer progression. The expression of BIRC7 was not consistently correlated with high stage in the three RCC groups (Figure 1g).

### 2.2. Evaluation of Function by BIRC5 Knockdown in RCC Cells

To evaluate the function of BIRC5 in RCC cells, a loss of function test was performed using siRNA for BIRC5. Caki-1, 786-O, and A498 ccRCC cell lines, Caki-2 and ACHN pRCC cell lines, and the UOK-276 chRCC cell line were used.

Reduced mRNA expression of BIRC5 was confirmed after siRNA-mediated BIRC5 knockdown in Caki-1 (ccRCC) and ACHN (pRCC) by qRT-PCR (Figure 2a). In addition, to support the universality of the BIRC5 knockdown effect in all RCC cells, the BIRC5 knockdown effect was also confirmed by evaluating BIRC5 mRNA levels in the second ccRCC cell line Caki-2 and the second pRCC cell line A498 (Appendix A). In cells with loss of BIRC5, cell viability was significantly reduced (11.2% of Caki-1, 10.8% of 786-O, 32.6% of A498, 28.4% of Caki-2, 22.6% of ACHN, and 33.7% of UOK-276 cells were still viable) and apoptosis was increased (Caki-1: early/late apoptosis: 9.00%/0.51%, 786-O: 11.50%/0.36%, A498: 10.5%/0.23%, Caki-2: 12.6%/0.7%, ACHN: 10.6%/2.04%, UOK-276: 1.81%/1.48%, respectively) (Figure 2b–d). We also performed wound healing and invasion assays. BIRC5 knockdown suppressed recovery of the scratched gap and invasive capacity of the RCC cell lines (Figure 2e–g). Therefore, inhibition of BIRC5 results in anti-cancer effects in cell lines of the three RCC subtypes.

### 2.3. Evaluation of Function by BIRC5 Overexpression in RCC Cells

To confirm the oncogenic effect of BIRC5 in RCCs, we performed a gain of function study using a BIRC5 overexpression plasmid. Through qRT-PCR, we confirmed that the expression of BIRC5 was increased 10,000-fold or more by transfection of the BIRC5 overexpression vector in Caki-1, ACHN, and UOK-276 cells compared with empty vector (EV) transfection (Figure 3a). In HK-2 cells, normal renal proximal tubular cells, cell viability was significantly increased by overexpression of BIRC5 (Figure 3a). BIRC5 induced an increase in cell viability of four cell lines (Figure 3b). In all cell lines, overexpression of BIRC5 led to reduced apoptosis compared with EV cells (EV and BIRC5 overexpression: 3.02% and 0.92% for Caki-1; 0.85% and 0.16% for ACHN; 0.83% and 0.26% for UOK-276; and 2.406% and 1.129 for HK-2, respectively) (Figure 3c,d).

We further evaluated the effect of overexpressed BIRC5 on cell migration and invasion. BIRC5 overexpression slightly increased the wound healing and invasive capacity of Caki-1, ACHN, UOK-276, and HK-2 cells compared with EV transfection (Figure 3e–g). Thus, overexpression of BIRC5 caused cancer-promoting effects in the three types of RCC cells and normal kidney cells.

### 2.4. Confirmation of Anti-Cancer Effect by BIRC5 Inhibition Using YM155

To investigate the potential therapeutic effect of inhibition of BIRC5, we tested the effect of the BIRC5 inhibitor YM155, a drug that increases apoptosis by inducing cleavage of caspase-3 suppressed by BIRC5 [16]. YM155 dose-dependently decreased cell viability in three RCC cell lines and normal kidney cells (Figure 4a). In response to treatment with 50 nM YM155, cell viability was effectively reduced in Caki-1 cells by 77%, ACHN cells by 78%, and UOK-276 cells by 75%. HK-2 cells showed a lower response, with reduction of 33%. Next, we investigated whether apoptosis was induced by YM155 treatment for 24 h. In all three RCC cell lines, apoptosis was promoted by YM155 treatment (0 nM and 50 nM: 7.76% and 48.38% for Caki-1 cells; 2.12% and 20.37% for ACHN cells; and 1.68% and 7.68% for UOK-276 cells, respectively) (Figure 4b,c). YM155 also inhibited both migration and invasion ability of RCC cell lines (Figure 4d,e). Therefore, these results indicate that inhibition of BIRC5 using YM155 induces cancer-specific anti-cancer effects for ccRCC, pRCC, and chRCC in vitro.

### 2.5. Identification of the Mechanism Underlying the Antitumor Effect of YM155

To examine the mechanism of the effects of YM155 treatment more closely in the three types of RCCs, we first examined the mRNA levels of BIRC5. The level of BIRC5 mRNA was clearly inhibited by YM155 in RCC (Figure 5a). Thus, we hypothesized that YM155 may epigenetically regulate the promoter region of BIRC5. To investigate acetylation, which is involved in gene regulation, we analyzed the genome browser database from University of California Santa Cruz (UCSC) genomic institute. In the UCSC genomic data analysis, we found that acetylated lysine 27 of histone H3 (H3K27Ac) was highly enriched in the promoter region of BIRC5 (Figure 5b). To examine whether YM155 influenced H3K27Ac enrichment of the BIRC5 promoter, we performed a chromatin-immunoprecipitation (ChIP) assay and found that YM155 decreased the enrichment of H3K27Ac in the BIRC5 promoter in all three cell lines (Figure 5c). We then speculated that the mRNA expression of EP300, an acetyltransferase that catalyzes H3K27Ac, may be affected by YM155 treatment. Consistent with our hypothesis, RT-PCR data showed that the mRNA expression of EP300 was decreased by YM155 (Figure 5d). These findings suggest that YM155 may control the expression of BIRC5 through regulating the expression of the EP300 acetyltransferase.

### 2.6. Confirmation of Antitumor Effect of YM155 on RCC Cells In Vivo

To confirm the effect of YM155 treatment on RCC cells in vivo, a subcutaneous xenograft tumor model was established by inoculating Caki-1, ACHN, and UOK-276 cells into athymic nude mice. When the tumor size reached 300 mm^3^, YM155 (5 mg/kg) was injected intraperitoneally, and the tumor size was measured for 4 weeks. As shown in Figure 6a, YM155 significantly reduced tumor growth of tumors derived from the three RCC cell lines in comparison with the control group (Appendix A).

Immunohistochemistry showed that P300 and H3K27Ac protein expression in xenografted tumors was reduced by YM155 treatment (percent of P300-positive areas in Caki-1: Con/YM155: 42.1%/9.9%, ACHN: 37.6%/14.0%, UOK-276: 29.3%/12.8%; percent of H3K27Ac-positive areas in Caki-1: Con/YM155: 66.8%/26.0%, ACHN: 51.0%/13.6%, UOK-276: 70.9%/36.9%; Figure 6b). Therefore, these findings suggest that BIRC5 exerts a cancer-promoting function in the three RCC subtypes and YM155 effectively suppressed three types of RCC by inhibiting histone acetylation at the BIRC5 promoter.

## 3. Discussion

Most studies on RCC have focused on the treatment options of ccRCC, as more than two-thirds of patients with metastatic renal cancer have ccRCC. However, treatment methods for pRCC and chRCC, which account for 15% of all RCCs, are limited. To overcome this limitation, we analyzed TCGA data for ccRCC, pRCC, and chRCC and found that BIRC5 was significantly increased in tumors of the three major tumor subtypes compared with non-tumor tissues and that elevated expression correlated with tumor progression. Loss- and gain-of-function studies for BIRC5 showed that BIRC5 regulates growth, apoptosis, and migration of RCCs. The therapeutic effect of YM155, a BIRC5 inhibitor, has been shown in several cancers; however, the effect and underlying mechanism in RCC subtypes were not previously understood. We investigated whether YM155 had an inhibitory effect in the three types of RCC. Anti-cancer effects of YM155 were also verified in cell lines and xenograft models derived from cells of the three types of RCCs. Additionally, since the epigenetic control is at the top of gene expression, we investigated what epigenetic alteration occurred by YM155. According to the UCSC database, H3K27Ac was highly enriched in the BIRC5 promoter region. We examined the level of P300, an acetyltransferase, and found that YM155 treatment decreased mRNA expression of P300. Additionally, the expressions of P300 and H3K27Ac were effectively suppressed by YM155 in xenografted tissues (Figure 7).

The mRNA expression of BIRC7 showed significantly higher expression in tumors compared to non-tumor in the three RCC subtypes (Figure 1e,f). However, BIRC7 had no correlation with tumor grade (Figure 1g). Several studies investigated BIRC7 as a potential therapeutic target for XX. Kitamura et al. reported that positive expression in only 60% of all RCC patients, so BIRC7 is not attractive as a target, but since it is recognized in blood, it may be an immune therapy target [28]. In addition, Chen et al. discovered that the long non-coding RNA CCAT1 as an upstream factor of BIRC7 and BIRC7 can regulate RCC cell viability and apoptosis by interacting with CCAT [29]. Various studies have identified the possibility of BIRC7 as a therapeutic target, but more research is needed on whether it can be a practical target in RCC.

BIRC5 is a member of IAP that participates in cell cycle regulation, inhibition of apoptosis, promotion of angiogenesis, and other biological effects, and BIRC5 is involved in the occurrence and development of tumors along with vimentin and p53. In particular, the expression of vimentin and BIRC5 showed higher expression levels in RCC compared to adjacent normal kidney tissue [30,31]. In RCC patients, BIRC5 has been identified as a prognostic biomarker for RCC survival in several studies evaluating the association between candidate gene expression and overall survival. The expression of BIRC5 was correlated with pathological grade and clinical stage and was also associated with the progression and poor survival of RCC patients [32,33,34,35]. Recently, a study examining the early diagnosis and prognostic value of BIRC5 in ccRCC based on Cancer Genome Atlas database and Human Protein Atlas database also showed that BIRC5 expression was significantly higher in ccRCC than in normal kidney tissue and correlated with the clinical stage and pathologic grade of ccRCC [36]. Therefore, high expression inhibition of BIRC5 is an effective indicator for predicting the poor prognosis of patients in RCC and may affect the overall survival rate of patients. BIRC5 inhibits apoptosis by blocking caspase activation and the main effect of YM155 is to induce apoptosis. Recent studies have reported the various mechanisms of YM155. YM155 induces apoptosis by autophagy-dependent DNA damage, not apoptosis that simply interferes inactivation of caspase in lung, breast, and prostate cancer [37,38,39]. In addition, it plays a role as a key regulator of mitosis and the transition from G2M phase to G1 in the cell cycle and the transition from G1 to S phase [40]. YM155 thus causes G1 or G2M arrest [41]. Kuo et al. found that the expression of BIRC5 and its upstream regulator, OX40, was increased in HIV patients, and YM155 selectively reduced HIV-infected CD4+ T cells, thus supporting the long-term survival of infected cells [42]. Carew et al. explored several strategies for therapeutic inhibition of BIRC5/Survivin function in RCC. Treatment with temsirolimus, an mTOR inhibitor, reduced BIRC5/Survivin levels partially; however, the level of BIRC/Survivin was increased by further treatment of YM155. YM155 has been confirmed to significantly improve anticancer activity in vitro RCC cell line panels and *in vivo* xenotransplantation models [43]. Sim et al. verified that in RCC cells, YM155 reduced the nuclear levels of p65 and phosphorylated p65 and attenuated the transcriptional ability of p65/p50 heterodimers. YM155 reduced the transcription of the NF-κB target gene BIRC5/Survivin, resulting in anticancer therapeutic effects. The ability of YM155 to block major transcriptional pathways, such as NF-κB, would have a significant impact on pharmacological effects [44].

Reduction of P300 by YM155 can inhibit acetylation in the promoter region of other genes as well as BIRC5, and the transcription of these genes might also be inhibited. In addition, since H3K27Ac is an active enhancer marker [45], it is possible to discover an important DNA regulatory region for regulating the expression of downstream targets of BIRC5, so further study on wide genomic regions is required. In addition, H3K27Ac was enriched in the BIRC5 promoter region, and there was a CpG island [46]. Most CpG islands exist in promoter regions and suppress gene expression [47,48]. However, in RCC, since CpG is in a suppressive state, it is presumed that overexpression of BIRC5 is maintained. While our results illustrate the mechanism of YM155 by P300-H3K27Ac enrichment, a study examining the selective increase of DNA methylation as another way to control the expression of BIRC5 is also needed.

With the discovery of diagnostic markers for various cancers, including prostate cancer, liver cancer, colon cancer, pancreatic cancer, and ovarian cancer, the promise of strategies for the early diagnosis of cancer is increasing [49,50,51,52,53]. However, the heterogeneity of RCC is the biggest obstacle to the development of diagnostic markers. The representative genetic mutation of RCC is VHL mutation, but this mutation is limited to ccRCC and is not common to pRCC and chRCC. While pRCC is associated with mutations for cMET and fumarate hydratase, these mutations are not related to ccRCC [54,55,56,57]. In addition, in the case of RCC, except for three types of RCC, it is very difficult to analyze the patient’s pathology because these cases are very rare. However, a diagnostic marker that would target 90% of all RCCs (ccRCC; 75%, pRCC; 10%, chRCC; 5%) would enable the early diagnosis of numerous patients and increase patient survival rate, so the development of a diagnostic marker is essential.

In this study, our results suggest BIRC5 as a key factor that can target ccRCC, pRCC, and chRCC. Our findings demonstrated the anti-tumor effect and epigenetic mechanism of YM155, a BIRC5 inhibitor, through in vitro and in vivo experiments. We believe that YM155 might be a therapeutic strategy for the three major subtypes of RCC.

## 4. Materials and Methods

### 4.1. Cell Culture

The human RCC cell lines Caki-1, Caki-2, 786-O, A498, and ACHN were purchased from Korean Cell Line Bank (Seoul, Republic of Korea). The immortalized human chRCC cell line UOK-276 was kindly provided by Dr. W. Marston Linehan (National Cancer Institute, Bethesda, MD, USA). Cell lines were cultured in RPMI-1640 or DMEM (Sigma-Aldrich, St. Louis, MO, USA) containing 10% fetal bovine serum (FBS; Sigma-Aldrich) and 1% antibiotic-antimycotic (Thermo-Fisher Scientific, Waltham, MA, USA). All cells were maintained in an incubator designed to maintain a 5% CO_2_ atmosphere, constant temperature, and humidity suitable for cell growth.

### 4.2. siRNA Transfection

RCC cells in 6-well plates were transiently transfected with 100 nmol/mL siRNA (Bioneer, Daejeon, Republic of Korea) using Turbofect Reagent from Thermo Fisher Scientific (Waltham, MA, USA). At 16 h after transfection, Opti-MEM was changed with growth media.

### 4.3. Transfection of Plasmids into Cells

Cells were seeded to an even cell density and grown to 70% confluency in 6-well cell culture plates and then and then transfected with empty pcDNA 3.1+ C-DYK vector and a vector containing cDNA of human BIRC5 (pcDNA3.1+ C-DYK BIRC5, GenScript, #OHu23310D) using Turbofect Reagent.

### 4.4. RNA Extraction and Reverse Transcription PCR

Total RNA was extracted from cells using TRIzol reagent, and AccuPower RT Premix (Bioneer, Daejeon, Republic of Korea) was used to perform reverse transcription of the extracted RNA. For qRT-PCR for amplification of cDNA, a LightCycler^®^ 480II (Roche, Basel, Switzerland) and the LightCycler^®^ 480 SYBR Green I Master (Roche, #04887352001) were used following the manufacturer’s recommendations. Amplification of cDNA was performed using the gene-specific primers listed in Appendix A.

### 4.5. Chomatin Immunoprecipitation (ChIP)

ChIP assays were performed using the Pierce™ Agarose ChIP kit (Thermo Fisher, #26156). following the instructions from the manufacturer. Briefly, 50 μg DNA fragmented to 200–500 bp in size by enzymatic digestion was pre-cleared with protein A magnetic beads. The pre-cleared DNA was immunoprecipitated with H3K27Ac (Abcam, #ab4729). DNA extraction was performed for chromatin fragments, and qPCR was performed on the DNA bound to each target protein. Immunoglobin G (IgG) was used as the negative control. The qPCR results were calculated as IP/1% input as follows: (IP − IgG)/(Input − IgG). ChIP primers are listed in Appendix A.

### 4.6. Wound Healing Assay

Cells were seeded to an even cell density and grown to 80% confluency in 6-well cell culture plates. After overnight incubation in a serum-free medium, cell layers were scratched with sterile yellow tips. Images for the width of the initial gap (0 h) and the residual gap at 24 h after wounding were captured and measured using an optical microscope at 40× magnification (Olympus, Shinjuku, Japan). For scratch image analysis, the average distance was manually collected by measuring 4 spots between the edges of the gap. Data are presented as mean ± SD. Three replicates were included in the analysis and an unpaired Student’s *t*-test was performed. Significance was considered at *p* < 0.05.

### 4.7. Invasion Assay

Cell motility was measured using a transwell chamber (BD Biosciences). The upper surface of the transwell chamber was coated with 100 μL Matrigel (0.3 mg/mL, BD Biosciences). After incubation for 3 h at 37 °C, the supernatant was removed; a 500 μL cell suspension was placed in the top chamber; as a chemoattractant, 700 μL 20% FBS medium was added to the bottom chamber. The chambers were placed in a 24-well plate and incubated overnight. The next day, the membrane was stained using 0.4% crystal violet. Images of the invaded cells were examined using a microscope at 40× magnification, and the number of cells were quantitated.

### 4.8. Annexin V/Propidium Iodide Staining Assay

The apoptosis rate was assessed using an Annexin V–fluorescein isothiocyanate (FITC) apoptosis detection kit (BD Biosciences, #556547, Franklin Lakes, NJ, USA). Cells were transfected with siBIRC5 and treated with various concentrations of YM155 for a further period of 24 h. The supernatant and trypsinized cells were rinsed in Dulbecco’s phosphate-buffered saline and suspended in 1× binding buffer at a concentration of 1 × 10^6^ cells/mL. Then, 5 μL FITC-conjugated Annexin V and 2 μL propidium iodide were added to 100 μL containing 1 × 10^5^ cells and incubated for 15 min at room temperature in the dark. After incubation, 400 µL binding buffer (1×) was added to each tube, and the cells were analyzed using a FACSCanto flow cytometer (BD Biosciences).

### 4.9. In Vivo Tumor Growth Experiment

BALB/c nude mice (female, 4 weeks old, 20 g) were purchased from Orient Bio and raised under appropriate conditions. Caki-1, ACHN, and UOK-276 cells (5 × 10^6^ cells/0.1 mL Hank’s Balanced Salt Solution) were injected subcutaneously in the left flank. YM155 was administered intraperitoneally 3 times per week for 4 weeks. The tumor sizes were measured every 2 to 3 days by a digital caliper, and tumor volumes were calculated using the formula volume = π/6 (length × width^2^). All experimental procedures were performed in accordance with the institutional guidelines approved by the Hanyang University Institutional Animal Care and Use Committee (approval number: 2020-0024A). All procedures related to the *in vivo* experiments and animal care were carried out in accordance with the approved guidelines. The study is compliant with the ARRIVE guidelines 2.0.

### 4.10. Immunohistochemistry

Paraffin-embedded tumor tissue specimens were sliced into 3-μm thick sections and mounted onto slides. The slide of each section was deparaffinized with xylene and hydrated by alcohol. The slide was washed in 0.6% H_2_O_2_ in methanol for 15 min then, the slides were treated with 0.1% Triton-X 100, washed three times with phosphate-buffered saline, and blocked with 10% goat serum. To examine the expressions of P300 (1:100, Invitrogen, #MA1-16622) and H3K27Ac (1:100) the slides were treated with a corresponding primary antibody overnight at 4 °C, and the next day, were incubated with a secondary antibody at room temperature. After staining with diaminobenzidine (DAB), the tumor sections were visualized under an optical microscope (Leica DM s00B microscope, Leica Microsystems Inc., Bufalo Grove, IL, USA) at ×200 magnification. Using software LEICA APPLICATION SUITE version 4.1.0, the stained area relative to the area of total tissue was calculated and quantified as a percentage.

### 4.11. The Cancer Genome Atlas (TCGA) Data Analysis

To investigate the expression of IAP genes including BIRC5 in a large cohort for RCCs, we obtained RNAseq-based gene expression profiling data from The Cancer Genome Atlas (TCGA) renal cell carcinoma project (KIRC; ccRCC, KIRP; pRCC, and KICH; chRCC). A log2 transformation for raw data of RNAseq was applied. The fold change for expression of the tumor and the normal tissues was compared.

### 4.12. Statistical Analysis

The results are expressed as the mean ± SEM. Most statistical comparisons were calculated by one-way ANOVA followed by Bonferroni’s post hoc test using Prism 7 (GraphPad, San Diego, CA, USA) to compare groups. *p* < 0.05 was considered to be statistically significant.

## Figures and Tables

**Figure 1 ijms-25-00216-f001:**
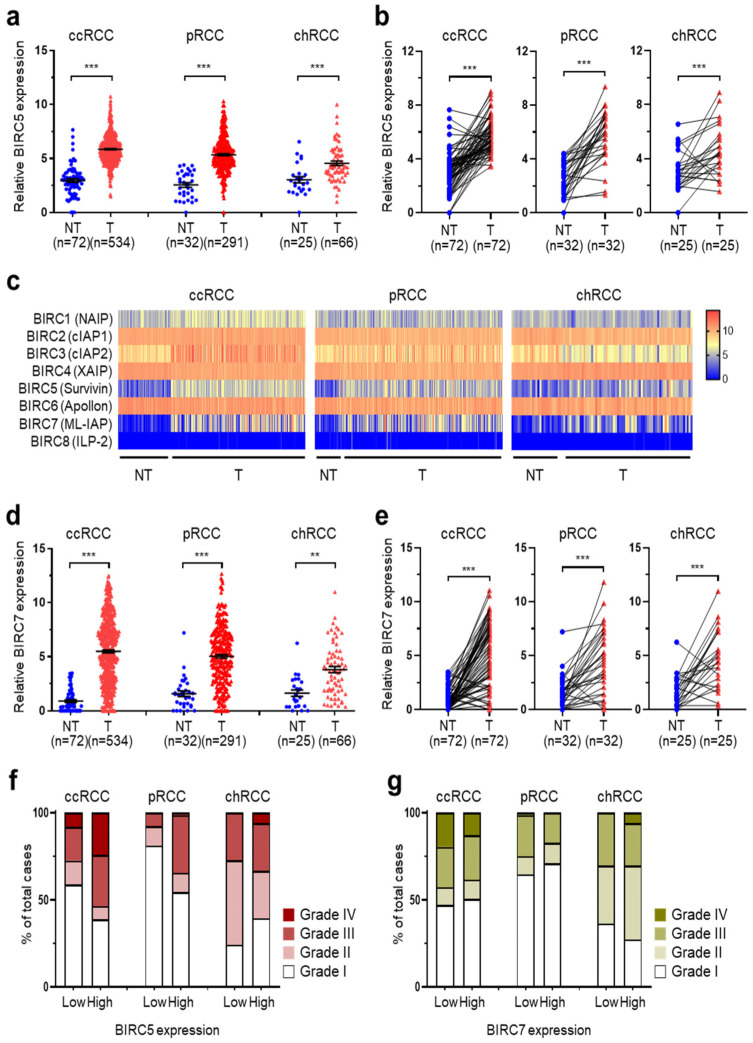
Overexpressed BIRC5 is associated with tumor progression in ccRCC, pRCC, and chRCC. (**a**) The relative BIRC5 mRNA expression levels in non-tumor (NT) and tumor (T) in TCGA datasets. The median expression level is indicated by horizontal lines (mean ± SD; *** *p* < 0.001 vs. non-tumor). (**b**) mRNA expression of BIRC5 in matched pairs of TCGA data. Data are shown as mean ± SEM. *** *p* < 0.001 by paired Student’s *t*-test. (**c**) Heat map of gene expression profiling for BIRC family in TCGA data for the three types of RCCs. (**d**) The relative BIRC7 mRNA expression levels in NT and T in TCGA datasets. The median expression level is indicated by horizontal lines (mean ± SD; *** *p* < 0.001 and ** *p* < 0.01 vs. NT). (**e**) mRNA expression of BIRC7 in matched pairs of TCGA data (mean ± SEM; *** *p* < 0.001 vs. NT). (**f**,**g**) Distribution of the clinical stages in RCCs with low and high (**f**) BIRC5 and (**g**) BIRC7 mRNA expression.

**Figure 2 ijms-25-00216-f002:**
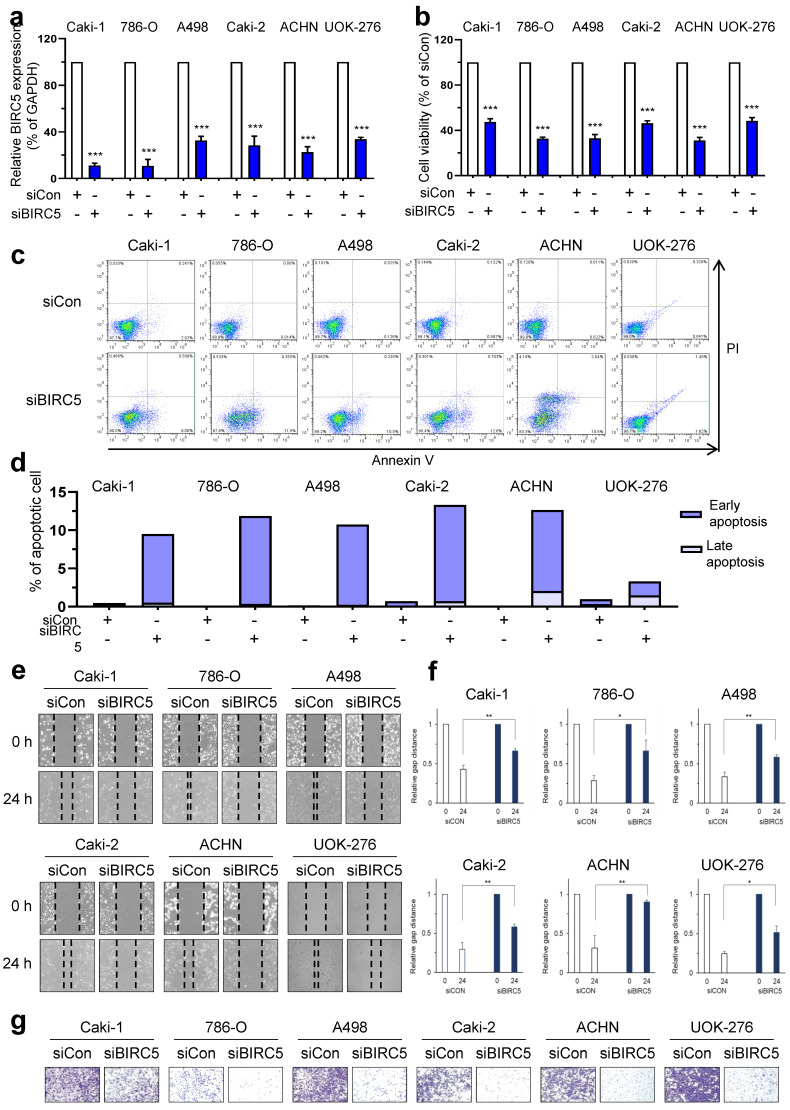
Loss of BIRC5 induced anti-cancer effect in ccRCC, pRCC, and chRCC cells. (**a**) BIRC5 mRNA levels were evaluated by qRT-PCR in Caki-1, 786-O, A498, Caki-2, ACHN, and UOK-276 cells transfected with the indicated siRNAs. Data are presented as the mean ± S.E.M. (*** *p* < 0.001 versus siCon). (**b**) Relative cell viability data are presented as the mean ± S.E.M (*** *p* < 0.001). (**c**,**d**) Annexin V-FITC/PI staining assay and representative images for FACS analysis were performed in BIRC5 knockdown Caki-1, 786-O, A498, Caki-2, ACHN, and UOK-276 cells. (**e**,**f**) Wound healing analysis after BIRC5 knockdown in Caki-1, 786-O, A498, Caki-2, ACHN, and UOK-276 cells. Data represent mean ± SD of three independent experiments (* *p* < 0.05 and ** *p* < 0.01 versus siCon, *t*-test). (**g**) Representative images of invasion assay results.

**Figure 3 ijms-25-00216-f003:**
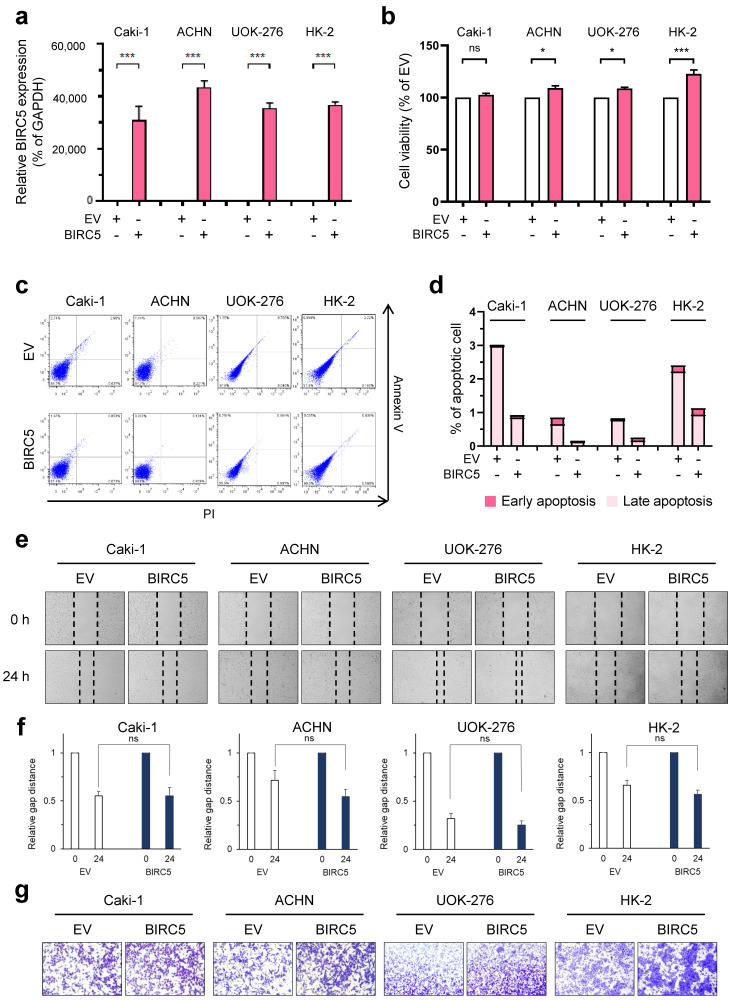
Ectopic overexpression of BIRC5 promoted the aggressive behavior of RCC cells. (**a**) BIRC5 mRNA levels were determined by qRT-PCR in Caki-1, ACHN, UOK-276, and HK-2 (normal renal proximal tubular cells) transfected with the indicated overexpression plasmids. Data are presented as the mean ± S.E.M. (*** *p* < 0.001 versus empty vector). (**b**) Relative cell viability data are presented as the mean ± S.E.M (ns; non—significant, *** *p* < 0.001 and * *p* < 0.05 versus empty). (**c**,**d**) Annexin V-FITC/PI staining assay and representative images for FACS analysis in BIRC5 overexpression Caki-1, ACHN, UOK-276, and HK-2 cells. (**e**,**f**) Wound healing analysis in Caki-1, ACHN, UOK-276, and HK-2 cells transfected as indicated. Graphs are shown to represent the mean represent mean ± SD of three independent experiments (ns; non—significant). (**g**) Representative images of invasion assay results.

**Figure 4 ijms-25-00216-f004:**
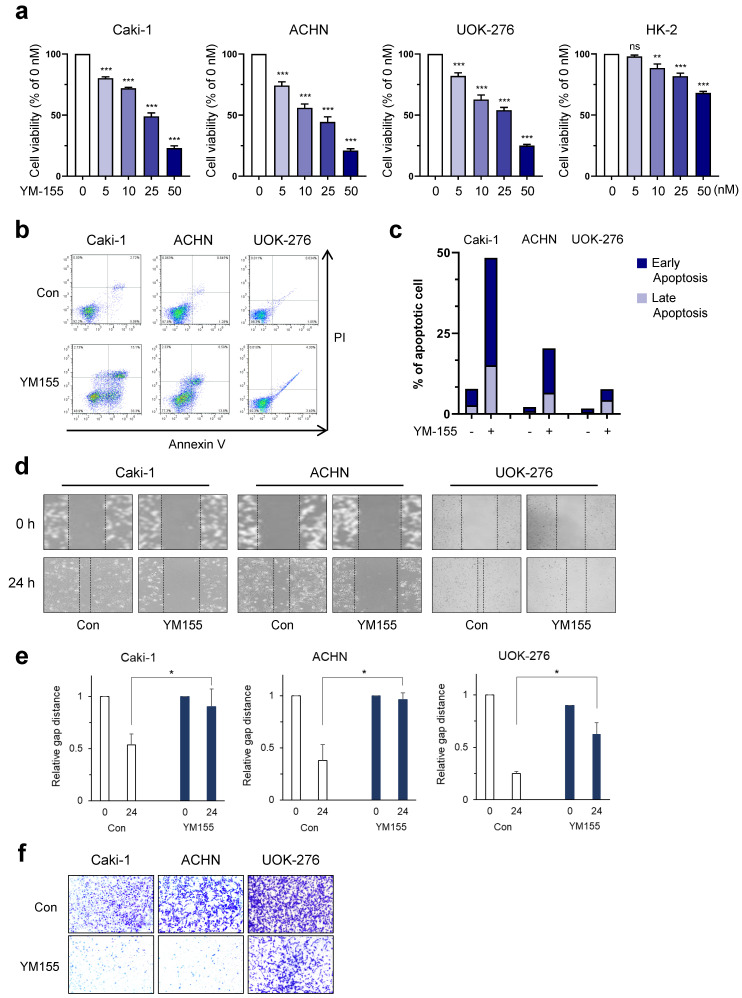
YM155 induced cancer-specific inhibition of cell growth in threetypes of RCCs. (**a**) Cell viability assay in HK-2, Caki-1, ACHN, UOK-276, and HK-2 cells. Cells were treated with increasing doses of YM155 (ns; non—significant, ** *p* < 0.01, *** *p* < 0.001 versus 0 nM). (**b**,**c**) Annexin V-FITC/PI staining assay and representative images for FACS analysis in Caki-1, ACHN, and UOK-276 cells treated with 50 nM YM155 for 24 h. (**d**,**e**) Wound-healing analysis in Caki-1, ACHN, and UOK-276 cells after treatment with YM155. Data represent mean ± SD of three independent experiments (* *p* < 0.05 compared to control, *t*-test). (**f**) Representative images of invasion assay results.

**Figure 5 ijms-25-00216-f005:**
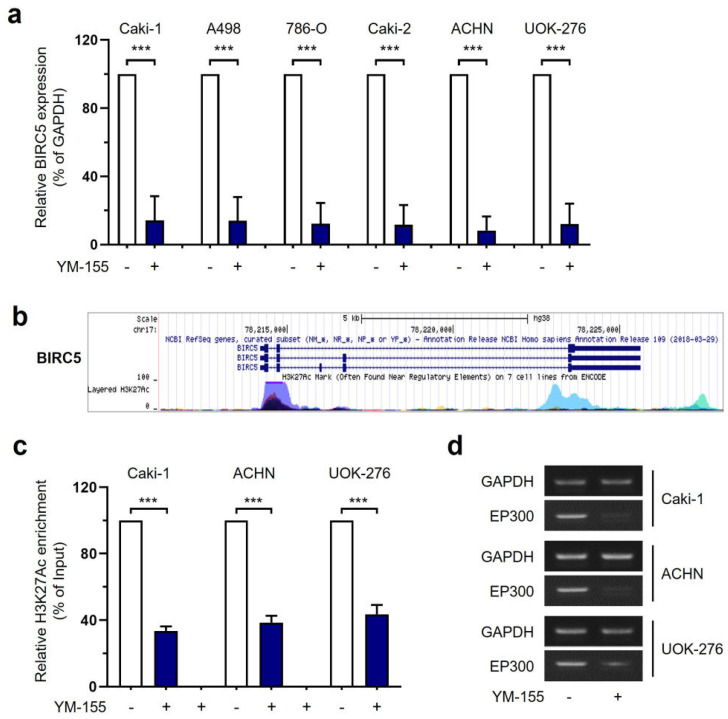
YM155 epigenetically interfered with histone acetylation in the promoter of BIRC5 and inhibited its expression. (**a**) BIRC5 mRNA levels were determined by qRT-PCR in 6 RCC cell lines after the treatment of YM155. Data are presented as the mean ± S.E.M. (*** *p* < 0.001 versus empty vector). (**b**) The promoter region of BIRC5 obtained from the UCSC database. H3K27Ac Mark (often found near active regulatory elements) on 7 cell lines from ENCODE. Navy blue; K562, sky blue; HUVEC, pink; NHLF, purple; NHEK, green; HSMM, yellow; H1-hESC, Orange; GM12878. (**c**) ChIP assays in Caki-1, ACHN, and UOK-276 cells. Crosslinked complexes were immunoprecipitated with anti-H3K27Ac antibody, followed by qPCR with the BIRC5-specific primers (mean ± SEM; *** *p* < 0.001 compared to control). (**d**) RT-PCR of EP300 and GAPDH mRNA levels in Caki-1, ACHN, and UOK-276 cells.

**Figure 6 ijms-25-00216-f006:**
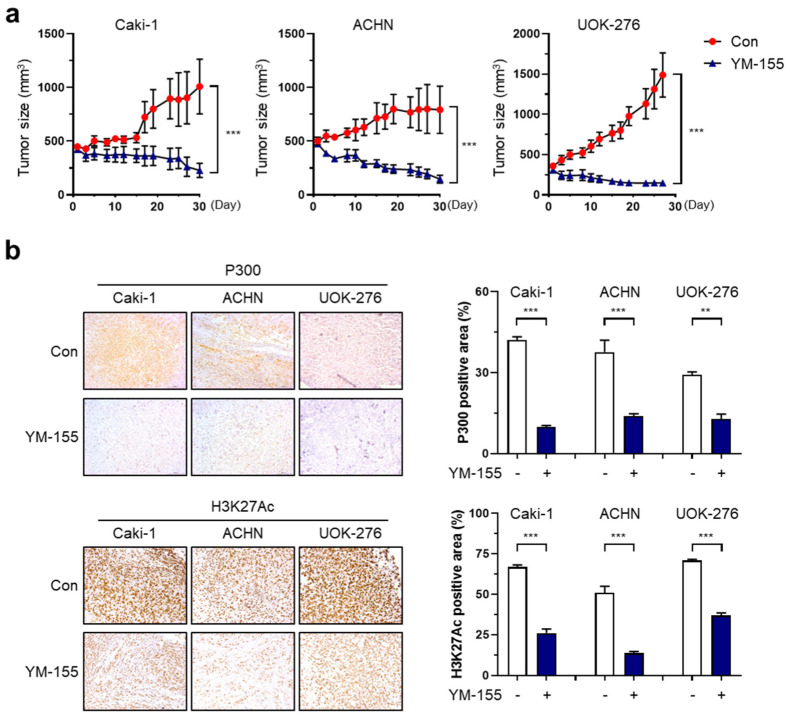
YM155 suppressed tumor growth through repressing levels of P300 and H3K27Ac in the three types of RCCs. (**a**) Xenografted tumor size was measured in BALB/c mice injected with cells and subjected to YM155 treatment. Data are presented as the mean ± S.E.M. (*** *p* < 0.001 vs. Con) (**b**) Immunohistochemical staining of P300 and H3K27Ac in xenograft tumors from BALB/c mice injected with Caki-1, ACHN, and UOK-276 cells and treated with YM155. *** *p* < 0.001 and ** *p* < 0.01 in unpaired *t*-test in the measurement of positive areas for P300 and H3K27Ac.

**Figure 7 ijms-25-00216-f007:**
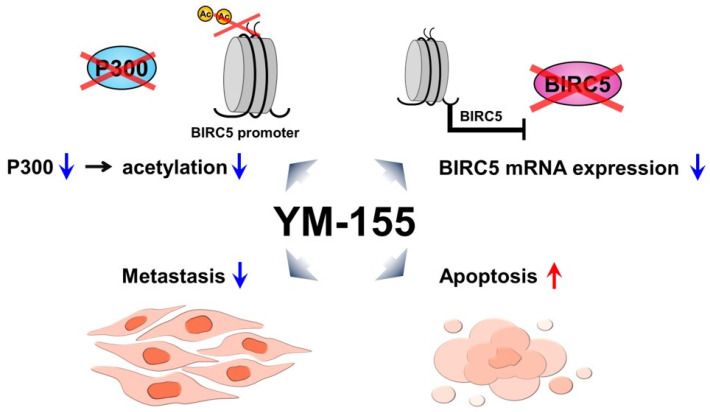
Schematic model for epigenetic regulation of YM155. BIRC5, which is abundant in three subtypes of RCCs, inhibits apoptosis of RCCs. YM155 inhibits BIRC5 by inhibiting an epigenetic enzyme and induces apoptosis in the three major subtypes of RCCs (blue arrow; suppression, red arrow; induction).

## Data Availability

The data that supports the findings of this study are available within the article.

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
