# Peer review of "Therapeutic Efficacy of YM155 to Regulate an Epigenetic Enzyme in Major Subtypes of RCC"

_ijms, 2023, doi:10.3390/ijms25010216_

Round 1
Reviewer 1 Report
Comments and Suggestions for Authors
The manuscript is well written but I have some major concerns regarding its novelty and impact since YM155 as been widely studied in RCC over the past years.
In the abstract section (and throughout the manuscript) the authors say that “In this study, we analyzed the public databases for ccRCC, 17 pRCC, and chRCC and found that BIRC5 was commonly overexpressed in a large cohort of pRCC and chRCC patients as well as ccRCC and was closely related to the progression of RCCs” . After a minor search online I found a paper of 2021 (doi: 10.1186/s13046-021-02026-1) were other authors described BIRC5 diffential expression among chRCC, ccRCC and pRCC using TCGA-RCC cohorts. Therefore this finding lacks novelty and was described years ago.
Moreover, YM155 is a well known and widely studied compound in RCC context over the years. In fact, there are already several published articles (using both cells lines and in vivo models) studying the combined or synergic effect of YM155 with other compounds in the treatment of RCC. Therefore the authors should include and discuss these data in the discussion section and also justify wh they chose to not combine YM155 with other compounds.
Here are some other studies that I found after a quick search online:
- in a paper from 2015 (doi: 10.1158/1535-7163.MCT-14-1036) the authors concluded that the BIRC5 suppressant YM155 exhibited promising activity in RCC models and augmented the in vitro and in vivo activity of temsirolimus through a mechanism characterized by cooperative depletion of BIRC5 levels;
- In a paper from 2017 (doi: 10.21873/anticanres.11291) the authors concluded that inhibition of survivin by YM155 overcomes statin resistance in RCC cells.
- In a paper from 2017 (doi: 10.1371/journal.pone.0178168) the authors investigated the mechanism of action of YM155 in the inhibition of B|IRC5 and RCC. These authors also tested YM15 in RCC cell lines and xenograft models and even combined it with sorafeniv to evaluate a potential synergic effect of therapies.
Minor editing of English language required
Author Response
1) The manuscript is well written but I have some major concerns regarding its novelty and impact since YM155 as been widely studied in RCC over the past years. In the abstract section (and throughout the manuscript) the authors say that “In this study, we analyzed the public databases for ccRCC, 17 pRCC, and chRCC and found that BIRC5 was commonly overexpressed in a large cohort of pRCC and chRCC patients as well as ccRCC and was closely related to the progression of RCCs”
After a minor search online I found a paper of 2021 (doi: 10.1186/s13046-021-02026-1) were other authors described BIRC5 diffential expression among chRCC, ccRCC and pRCC using TCGA-RCC cohorts. Therefore this finding lacks novelty and was described years ago. Moreover, YM155 is a well known and widely studied compound in RCC context over the years. In fact, there are already several published articles (using both cells lines and in vivo models) studying the combined or synergic effect of YM155 with other compounds in the treatment of RCC. Therefore the authors should include and discuss these data in the discussion section and also justify wh they chose to not combine YM155 with other compounds.
Here are some other studies that I found after a quick search online:
- in a paper from 2015 (doi: 10.1158/1535-7163.MCT-14-1036) the authors concluded that the BIRC5 suppressant YM155 exhibited promising activity in RCC models and augmented the in vitroand in vivo activity of temsirolimus through a mechanism characterized by cooperative depletion of BIRC5 levels;
- In a paper from 2017 (doi: 10.21873/anticanres.11291) the authors concluded that inhibition of survivin by YM155 overcomes statin resistance in RCC cells.
- In a paper from 2017 (doi: 10.1371/journal.pone.0178168) the authors investigated the mechanism of action of YM155 in the inhibition of B|IRC5 and RCC. These authors also tested YM15 in RCC cell lines and xenograft models and even combined it with sorafeniv to evaluate a potential synergic effect of therapies.
Answer: We appreciate the time and effort that reviewer 1 has invested in the detailed review of our manuscript. We have strived to refine our analyses and interpretations based on the guidance provided by reviewer 1. As suggested by the reviewer, we agree that published information exists regarding the efficacy of YM155 in inhibiting RCC and its synergistic interactions with other drugs over the past years. Similar to the article you indicated, we identified through public data that BIRC5 is possible biomarker for RCC. However, we went beyond this by demonstrating the potential ability of BIRC5 as a biomarker through in vitro experiments for three types of RCC cell lines and in vivo experiments. Furthermore, we not only revealed the anti-cancer effect of YM155 in three types of RCC but also elucidated the underlying mechanism that YM155 hinders the progression of RCC by reducing acetylation through inhibition of epigenetic enzymes. To our knowledge, no studies have addressed mechanism of YM155 on three major types of RCC, so we believe that our study is meaningful because it is the first to suggest the epigenetic mechanism of YM155 through in vitro and in vivo experiments.
"Attached file has been added."
Reviewer 2 Report
Comments and Suggestions for Authors
Hong and colleagues evaluated publicly available TCGA datasets for ccRCC, pRCC and chRCC and identified BIRC5 as highly expressed in all 3 RCC cohorts. BIRC5 expression correlated positively with tumor grade. Knockdown of BIRC5 in ccRCC pRCC and chRCC human renal cancer cell lines resulted in reduced cell viability, increased apoptosis, reduced migration and invasion in targeted assays. Conversely, BIRC5 overexpression in these cell lines led to increased cell viability, reduced apoptosis and slight increase in wound healing and invasive capacity. BIRC5 inhibitor YM155 treatment of representative RCC cell lines resulted in reduced cell viability, increased apoptosis, and inhibition of migration and invasion ability of RCC cell lines and functions by interfering with histone acetylation in the BIRC5 promoter. They suggest BIRC5 inhibition could be potential therapy for RCC.
The paper is well written and data convincing to support the conclusions. However, the authors did not do a thorough literature search on this topic. There are a number of papers that evaluate survivin as a prognostic marker in RCC and these should be referenced and discussed relative to this work.
Points to address:
1) Fig 2a should be preceded by a graph showing relative BIRC5 gene expression in the 6 cell lines relative to HK2 or RPTEC cells for context. Carew et al, Mol Can Ther (2015) 14:1404 evaluates survivin gene expression in 5 of 6 RCC cell lines used here and show high expression of survivin in these cell lines and that survivin knockdown results in inhibition of tumor growth in 786-0 xenografts.
2) Fig 3e-wound healing images are not very clear and one is not able to see cells. They should present an improved figure to show the result. Is there any way to quantitate this assay?
3)It would add support to the universality of the results for all RCC subtypes if experiments in Fig 3, Fig 4 and Fig 5c showing effects of BIRC5 knockdown in Caki-1 (ccRCC) and ACHN (pRCC) were repeated in a second ccRCC cell line and a second pRCC cell line.
4) Fig 3 legend needs a stated p value for the single asterisk.
5) Should cite and discuss papers documenting survivin as a pathological and prognostic marker in RCC (45 hits were found when BIRC5/survivin and RCC were used as search terms in PubMed): 1) Pu et al., Oncotarget (2017) 8:19825; 2) Berglund et al., Cancer Med (2020) 9:8662; and others
6) Should cite and discuss paper in which the survivin inhibitor YM155 was evaluated and shown to be effective as a therapeutic agent in RCC models. Carew et al, Mol Can Ther (2015) 14:1404.
7) Line 227-tissue progression should be tumor progression
Author Response
Comments and Suggestions for Authors 2
→ We really appreciated excellent critics from reviewer and tried our best to revise based on reviewer comments.
1) Fig 2a should be preceded by a graph showing relative BIRC5 gene expression in the 6 cell lines relative to HK2 or RPTEC cells for context. Carew et al, Mol Can Ther (2015) 14:1404 evaluates survivin gene expression in 5 of 6 RCC cell lines used here and show high expression of survivin in these cell lines and that survivin knockdown results in inhibition of tumor growth in 786-0 xenografts.
Answer: We agree with the reviewer's suggestion regarding the presentation of differential BIRC5 expression across 6 cell lines relative to HK2 or RPTEC cells. However, we have already demonstrated, through public data, an elevated expression of BIRC5 in tumors compared to non-tumor samples in our study (Figure 1a, 1b, 1c, and 1f). As you have indicated (Carew et al, Mol Can Ther (2015) 14:1404), mRNA expression among cell lines has already been widely addressed in several papers. Furthermore, given that in our research, the emphasis lies more on the sensitivity to YM155 than on the expression of BIRC5 (Figure 4a), we believe that repeating this information could be redundant for the reader.
2) Fig 3e-wound healing images are not very clear and one is not able to see cells. They should present an improved figure to show the result. Is there any way to quantitate this assay?
Answer: Thanks for your comments. As mentioned you, we presented the improved results graphically and performed statistical analysis (Fig 3e and 3f). In addition, the legend in Fig. 3 was revised. All wound healing assay data in Fig. 2e and Fig. 4d were also graphically expressed and subjected to statistical analysis (Fig. 2e and 2f, Fig. 4d and 4e). The legends for Fig. 2 and Fig. 4 have been modified. The analysis method was written in 4.6. Wound healing assay, lines 363~366.
3) It would add support to the universality of the results for all RCC subtypes if experiments in Fig 3, Fig 4 and Fig 5c showing effects of BIRC5 knockdown in Caki-1 (ccRCC) and ACHN (pRCC) were repeated in a second ccRCC cell line and a second pRCC cell line.
Answer: According to your comments, Caki-2 as the second ccRCC cell line and A498 as the second pRCC cell line, the BIRC5 knockdown effect was confirmed by qRT-PCR. Results were added to lines 119~122 by evaluating BIRC5 mRNA levels. Data was added as Supplementary figure 1.
4) Fig 3 legend needs a stated p value for the single asterisk.
Answer: P* < 0.05 versus empty was written in the legend to Fig. 3.
5) Should cite and discuss papers documenting survivin as a pathological and prognostic marker in RCC (45 hits were found when BIRC5/survivin and RCC were used as search terms in PubMed): 1) Pu et al., Oncotarget (2017) 8:19825; 2) Berglund et al., Cancer Med (2020) 9:8662; and others
Answer: Thank you for your advice. According to your comments, we cited and discussed papers documenting BIRC5/survivin as a pathological and prognostic marker for RCC in renal cell carcinoma (lines 263 ~277).
And, we have more added the references as follows;
- Shi, Z.G.; Li, S.Q.; Li, Z.J.; Zhu, X.J; Xu, P.; Liu, G. Clin Transl Oncol 2015, 17, 65–73.
- Oto, O.A.; Paydas, S.; Tanriverdi, K.; Seydaoglu, G. et al. Leuk Res 2007, 31, 1495–501.
- Berglund, A.; Amankwah, E.K.; Kim Y.C. et al. Cancer Med 2020, 9, 8662–8675.
- Ma C.; Lu B.; Sun E. Postgrad Med J 2017, 93, 186–192.
- Petitprez, F.; Ayadi, M.; Reyniès, A.; Fridman, W.H. et al. Front. Oncol 2021, 1–11.
- Pu, Z.; Wang, Q.; Xie, H.; Wang, G.; Hao, H. Oncotarget 2017, 8, 19825–19833.
- Wang, J.; Chen, M.; Dang, C.; Zhang, H.; Wang, X. et al. Urol Int 2022 106, 344–351.
6) Should cite and discuss paper in which the survivin inhibitor YM155 was evaluated and shown to be effective as a therapeutic agent in RCC models. Carew et al, Mol Can Ther (2015) 14:1404.
Answer: Per reviewer suggestions, the paper using YM155 as a treatment in the RCC model was cited and discussed, and the contents were added to lines 286 to 295. And, we have more added the references as follows;
- Jennifer S. Carew, J.S.; Claudia M. Espitia, C.M et al. Mol Cancer Ther; 2015 14, 1404–1413.
-
Sim, M. Y.; Shyi, J.S.; Yuen, P.; Go, M.G. Scientific Reports 2018 8, 10289.
7) Line 227-tissue progression should be tumor progression
Answer: According to your comments, we changed it to the word "tumor progression" on Line 235.
"An attached file containing the updated text of the manuscript has been added."
Round 2
Reviewer 1 Report
Comments and Suggestions for Authors
My questions have been answered. I have no further comments.
Reviewer 2 Report
Comments and Suggestions for Authors
The authors have addressed the comments of this reviewer in a satisfactory manner.